# Enhancing Behavior Change Skills in Health Extension Workers in Ethiopia: Evaluation of an Intervention to Improve Maternal and Infant Nutrition

**DOI:** 10.3390/nu13061995

**Published:** 2021-06-10

**Authors:** Vivien Swanson, Joanne Hart, Lucie Byrne-Davis, Rowena Merritt, Wendy Maltinsky

**Affiliations:** 1Psychology Division, School of Natural Sciences, University of Stirling, Stirling FK9 4LA, UK; wendy.maltinsky@stir.ac.uk; 2Division of Medical Education, University of Manchester, Oxford Rd, Manchester M13 9PT, UK; jo.hart@manchester.ac.uk (J.H.); lucie.byrne-davis@manchester.ac.uk (L.B.-D.); 3Centre for Health Services Studies, University of Kent, George Allan Wing, Canterbury CT2 7NF, UK; R.K.Merritt@kent.ac.uk

**Keywords:** food security, food intake, eating behaviour, low- and middle-income countries, infant nutrition, behavioral intervention, staff training, health extension workers

## Abstract

Maternal and infant nutrition are problematic in areas of Ethiopia. Health extension workers (HEWs) work in Ethiopia’s primary health care system, increasing potential health service coverage, particularly for women and children, providing an opportunity for health improvement. Their roles include improving maternal and infant nutrition, disease prevention, and health education. Supporting HEWs’ practice with ‘non-clinical’ skills in behavior change and health communication can improve effectiveness. This intervention study adapted and delivered a UK-developed training intervention for Health Extension Workers (HEWs) working with the United Nations World Food Programme in Ethiopia. The intervention included communication and behavioral training adapted with local contextual information. Mixed methods evaluation focused on participants’ reaction to training, knowledge, behavior change, and skills use. Overall, 98 HEWs were trained. The intervention was positively received by HEWs. Pre-post evaluations of communication and behavior change skills found a positive impact on HEW skills, knowledge, and motivation to use skills (all *p* < 0.001) to change women’s nutritional behavior, also demonstrated in role-play scenarios. The study offered substantial learning about intervention delivery. Appropriate cultural adaptation and careful consideration of assessment of psychological constructs are crucial for future delivery.

## 1. Introduction 

Nutritional intake in parts of Ethiopia can be challenging due to poor food accessibility, difficulties storing food, and limited food variance [1]. The consequences for young children are particularly problematic, resulting in stunting at high levels [2,3]. Pregnant women also require good nutritional intake to reduce risks of fetal deficits, yet consumption of dairy products, eggs, meat, fruits, and vegetables is suboptimal, resulting in low levels of calcium, iron, zinc, and folic acid [1]. Nutritional intake of pregnant women and mothers can also be impacted through cultural and religious practices and social norms around fasting, partner and family influences [4], as well as food taboos with beliefs, for example, that certain food groups might render the fetus too large for the birth canal [5]. 

Health extension workers (HEWs) work in communities on the frontline of Ethiopia’s primary health care system [6]. They are generally female secondary school graduates selected from their own communities and receive a one-year training in health service delivery. The introduction of HEWs has exponentially increased health service coverage from 64% of Ethiopia’s population in 2004 to 92% in 2011, providing an important source of health improvement skills and advice. Their main tasks include improving infant and maternal nutrition, disease prevention, and health education. One significant remit of Ethiopian health extension workers (HEWs) in the remote and poor areas of the country is mitigating high infant and maternal mortality. Support includes nutritional products and nutritional information and advice, which aim to improve nutritional behaviors at the community level [7,8]. Nevertheless, women do not consistently use the nutritional supplements provided, their own nutritional consumption often remains poor, and many do not act on the advice of the HEWs. 

A community based qualitative interview-based study carried out in the Amhara region of Ethiopia in 2017 on behalf of the World Food Programme (WFP) [9] suggested that although women report that they receive sufficient information about nutrition from HEWs, in many cases, they lack the skills of how to overcome contextual barriers to providing nutritional diets for themselves and their children. The WFP [9] study identified reasons such as perceived financial barriers, difficulties changing habitual or traditional ways of feeding children, and social pressure around fasting from families; barriers which could be addressed by using appropriate skills and techniques based on theories of behavior change [10,11]. HEWs often experience frustration surrounding their role, which relies on changing the behaviors and beliefs of women regarding dietary practices [6,9]. HEWs receive training in health care knowledge [12,13], and some may have received training in interpersonal counseling [14]; however, there is limited training on the use of behavior change techniques that help people change behaviors. 

Health educators are increasingly aware that health-related knowledge is necessary but not sufficient to achieve behavior change [15]. Behavior change skills used together with collaborative, person-centered communication skills can help people to initiate and maintain changes to health and lifestyle behaviors [16,17] by helping them explore reasons for their behavior and guide them to make more healthy choices. This approach is frequently used in the UK health services and internationally [18] but has not been used in the context of maternal nutrition behaviors in LMICs such as Ethiopia, and it is not clear how easily these skills and concepts transfer to an LMIC context [19]. To improve the effectiveness of health behavior change interventions, there is a need to support preparation for practice and performance with this type of additional ‘non-clinical’ skills and competencies for many health professionals internationally [20], and this is particularly appropriate for HEWs.

The Change Exchange [21] is a collaborative of UK and international health psychologists and other professionals who support global health partnerships/organizations to achieve behavior change in their work. This includes understanding barriers and facilitators to change, developing theories of change, designing or adapting interventions to include different behavior change techniques, and evaluating the impact of partnership work on behaviors. The Change Exchange supported the project in Ethiopia reported in this paper. 

### 1.1. Changing the Behavior of Health Professionals

Behavior change techniques (BCTs) are the smallest evidence-based components of a behavior change intervention [22]. BCTs can be used by health professionals to assist behavior change when used in conversations/consultations with individuals [23]. Training health professionals to use BCTs requires approaches that help them to understand BCTs’ value, believe that the outcomes of using the techniques are more favorable than current practice, feel competent in using them, and can plan how to use the techniques and how they may overcome barriers to their use in practice. 

There is evidence that sets of specific, defined behavior change techniques from the Health Behaviour Change Competency Framework [24] and elsewhere [22,23] are important for effective behavior change; however, deciding which techniques to use in which context can be complex. Those based on the MAP acronym (Motivation, Action, and Prompts) [24] have excellent theoretical and face validity, are clearly structured, and can be effective in changing health-related behaviors. It has been shown that it is possible to train health professionals in these techniques to help them help their clients to improve their health [18]. 

To use behavior change techniques effectively, person-centered communication skills are crucial [17]; for example, being able to listen to and understand people’s problems and reasons for not making changes and asking open questions and reflections to help guide them towards adopting more healthy choices. HEWs may find these skills useful to help families improve nutritional behaviors. 

### 1.2. Aims

The primary aim of this project was to develop, deliver and evaluate a program of communication and behavior change skills training for HEWs in the Amhara region of Ethiopia to support them to have effective behavior change conversations with mothers of infants and young children about nutrition. 

This included assessment of the feasibility of delivery, acceptability to participants, and participants’ learning, including an increase in knowledge, competence, and confidence to implement the techniques following training. The goal was to support HEWs in their role by enhancing their skills to help mothers provide better nutrition for their young children. This was part of a UN World Food Programme (WFP) project in Maternal and Infant Nutrition. The WFP engaged with The Change Exchange (see www.mcrimpsci.org; accessed on 24 February 2021) to provide behavior change expertise. 

## 2. Materials and Methods

### 2.1. Intervention Development

The training program was an adaptation of the MAP behavior change Programme developed in Scotland, based on the HBCCF [18,24]. Delivery plans, materials, and assessment tools were prepared in the UK in advance of intervention delivery. All teaching materials, including PowerPoint presentations, session plans, and course tutor notes, were reviewed by members of the WFP for cultural and workforce appropriateness. Recommended changes were discussed with the members of the Change Exchange, including UK-based specialists in behavior change (W.M. and V.S.), who would be delivering the training intervention. The materials were then translated into Amharic by the WFP as the first language in the region of Ethiopia. It was recognized that adjustments to the training would be needed immediately prior to and during delivery, based on information gathered and testing of approaches in the local context. For example, it was crucial to understand the specific attitudes, barriers, and cultural pressures on women preventing them from adopting healthier nutrition for their children. It was also important to understand the structure of local services, how this communication and behavior change approach might be embedded in services, and how the development of the new skills could be sustained and supported in this context. Regular meetings with local WFP representatives and the research team were held over the duration of the project, including orientation briefings at the WFP Ethiopia Country Office and on-site preparation prior to local delivery. Delivery of intervention workshop sessions in Ethiopia took place in April/May 2018. The intervention was delivered twice to cover the whole cohort. After the first two-day workshop (Group 1), the program was slightly modified for Group 2, where it was apparent that examples or methods were not effective or relevant for participants in this context. Modifications were minor and largely involved incorporating some of the examples that emerged from Group 1 to demonstrate to Group 2, making techniques specific to the context. For example, we focused on eating one egg as a goal as this was particularly relevant to all participants in Group 1. Information gathered from Group 1 indicated that the husband or mother could frequently be the person who could remind the HEW client to buy eggs, acting as both social support and as a prompt, so this example was used more frequently in the second delivery. Both groups of participants experienced the same training. Tailoring examples to fit participant context is good practice in any training delivery, but this did not change the behavior change techniques or communication skills used in training. The research team also participated in discussions and gathered information and feedback from WFP staff in the local area about the best way to tailor training to the local context, and participated in a Field Visit to a village where two HEWs worked to see their local clinic and observe their practice with clients. This provided very useful contextual information. 

### 2.2. Participants

The intervention was delivered to two groups of HEWs from three pilot areas in Ethiopia around Dessie: Kobo, Dessie Zuria, and Habru. A total of 98 health education workers participated in two workshops consisting of two full-days each (2 × 8 h meetings for each workshop): 57 in workshop 1 (Group 1) and 41 in workshop 2 (Group 2), delivered on 2 consecutive weekends. All were female and had been in their HEW role for an average of 8.6 years (ranging from zero to 15 years in post). Nearly three-quarters (62, 72%) had some previous experience of communication training. 

### 2.3. Intervention Delivery

The intervention was delivered in English, with one translator to Amharic, who was a specialist in nutrition with some experience in behavior change theory and research and augmented with translation support from local WFP staff. Materials included short PowerPoint presentions, written worksheets/handouts, flip charts for group work, pens, paper, and modeling material for imaginative exercises, and tablets for video recording of role-plays. All written materials were provided in Amharic. Methods of adult experiential learning were used [25], including modeling behaviors, short teaching sessions, games, active and imaginative exercises, simulations, role-plays, and group work. 

The intervention duration was two full days (8 h per day). It was delivered centrally in a conference room in the Amhara region, and most (95%) participants in Group 1 and Group 2 completed workshops on both days. 

### 2.4. Intervention Content (Groups 1 and 2) 

Day 1: Communication skills (open questions, affirmations, reflecting, and summarizing); identifying goals (what needs to change) and understanding barriers and facilitators.

Day 2: Motivational, action, prompting (MAP) behavior change techniques including setting goals, making plans and problem solving, understanding habits. 

Table 1 briefly describes the communication skills and MAP Techniques used in the training based on the HBCCF Taxonomy [24] (Table 1 here).

### 2.5. Measures 

It was important to evaluate the impact of the training intervention using a range of methods sensitive to cultural context. Kirkpatrick’s model was utilized [26]. This uses a 4 level hierarchical approach, including Level 1: evaluation of reaction to training; Level 2: learning achieved; Level 3: behavior changes achieved and Level 4: health outcomes attained. The first 3 stages of the model were used in this evaluation. We were unable to evaluate health outcomes due to resources and time. 

Methods included quantitative measurement using questionnaires pre- and post-delivery of the intervention, written open feedback from HEWs, and analysis of recorded transcripts of training to identify behavior change techniques (BCTs) utilized. 

### 2.6. Level 1: Reaction to Training and Learning Achieved (Post Training)

HEW Reactions to Group 1 workshops were not formally evaluated with participants. HEWs reactions in the Group 2 workshops (*n* = 41 participants) were evaluated using 9 items from the Training Acceptability Rating Scale (TARS) [27]. It measures improvements in understanding, confidence, and usability of skills, coverage of topics, satisfaction with the workshop, and leaders’ motivational skills and relationships. TARS items were scored from 1 = ‘not at all’ to 4 = ’a great deal’ and totaled to represent overall training satisfaction (possible range 9–36; alpha = 0.63). 

### 2.7. Level 2: Learning 

The TARS also includes two open feedback questions assessing learning from the workshop: What was the most helpful (knowledge and skills gained) part of the workshop for you personally?What change(s), if any, would you recommend? (e.g., to the content or teaching)

### 2.8. Level 3: Behavior Change

Readiness to change was assessed immediately pre- and post-training through a single multiple-choice questionnaire about HEW’s willingness to change the way they talked to women about nutrition. Five fixed-choice responses (scored 1–5) represented no willingness to complete willingness to change: e.g., ‘I don’t want to change…’ to ‘I have already started to change…’ 

Four behavior change techniques were measured using self-report immediately before training and at follow up after training and scored on a 0–10 scale with the root as follows: 

‘In your current work, for every 10 women you speak to about nutrition, how many of them do you………. to change the way they feed themselves and their babies ’

Items specified:Find out how much they already know before you talk to them.Talk about what problems they face.Help to make a plan to change.How many … Make the changes you suggest.

Post-training (one-month community-based follow-up), the same questionnaires were distributed and collected locally by WFP staff to record changes in HEWs work-related behavior. These were returned by mail to researchers in the UK and matched using unique identifiers for paired analysis. The analysis compared individual items before and after training. 

Some of the practical work in the training was audio or video recorded. Written feedback was translated by WFP staff. Feedback from local WFP staff regarding their observations of participants’ engagement and reactions to the training was discussed post-workshops. 

### 2.9. Role-Play Video Recordings (During Training)

Participants demonstrated the use of behavior change techniques in role-play scenarios. A selection of these in each workshop was video recorded using tablets with participants’ consent. Discussions in videos were translated from Amharic to English by local WFP translators and coded by researchers for observed use of behavior change techniques and communication skills. The role-play task invited HEWs to use the BCTs shown in Table 1, including goal setting, action planning, and coping planning (including problem-solving and decision making). We used a standard scenario based on relevant local issues to encourage mothers to provide their children with nutritious food—e.g., one egg per day. HEWs were encouraged to use open questions and reflective listening to set the conversation agenda (an outcome goal) and use MAP BCTs. 

### 2.10. Ethical Issues

All procedures contributing to this work complied with the ethical standards of the relevant national and institutional guidelines on human experimentation and with the Helsinki Declaration of 1975, as revised in 2008. The project was part of the ‘World Food Programme Social Behaviour Change Communication Project’ to improve local nutrition services and outcomes. Approval was gained from the local government before the training commenced. All data were collected in accordance with local and UK ethical principles and anonymized. Written consent for the use of anonymized data was obtained from all participants prior to participation in all components of the study. Anonymized data was stored on secure servers at the University of Stirling in accordance with the UK Data Protection Act, 2018. 

## 3. Results 

Engagement in the workshops was excellent, with most participants in Group 1 and Group 2 attending both workshop sessions. Follow-up data were obtained from 66 (68%) participants. Overall, training was very well received by participants in the HEW workshops. Written and oral feedback suggested they found it enjoyable, interactive, and interesting, for example: 

HEW, Group 1: “the training approach was very interactive and enjoyable. It was very good training and I wish that you repeat the training in the future”.

### 3.1. Reaction to Training

Total TARS scores were high (total mean = 34.7 (SD 1.5)). The lowest item score was 3.6 (SD.53) for coverage of intended topics and the highest was 3.9 (SD.15) for workshop leaders’ skills, suggesting workshops were enjoyed, valued, and highly rated by the HEWs. 

### 3.2. Helpful Knowledge and Skills Gained

Feedback also included written comments, categorized as ‘helpful knowledge and skills gained’ or ‘suggested improvements for the training’. Via written comments, HEWs reported improved knowledge about healthy nutrition (eating fruit, meat, eggs, milk, etc.) and the need for balanced diets acquired through discussion in the workshops, for example: 

HEW Group 1: “I have learnt to motivate and encourage mothers while discussing with them and to repeat the points they raised for themselves and to summarize at the end”.

HEW Group 2: “I have got a relevant knowledge about mothers and child feeding and how to communicate with mothers to bring behavioral change”.

They reported understanding the need for change in women’s and children’s nutritional behaviors and the need to change their own behavior and practice before they could support community or family behavior change. They acknowledged the important role of the health extension worker in behavior change; felt they had learned what qualities they needed to be a more effective HEW and make changes in community contexts. They reported gaining skills in communicating with mothers, listening, discussing effectively, and using open questions, and that their skills in motivating women through discussion were improved. 

#### Suggested Improvements

HEWs mentioned the length of training, with more time (more, longer sessions) needed to cover training adequately. More materials (worksheets) or a learning module to support training were requested. These included a request for training to be regularly repeated, reviewing progress, and reflecting on practice. To improve sustainability, training for senior and higher-level staff (who were largely English-speaking) was also seen as desirable, to offer local and managerial support, and for them to fulfill a coaching/mentoring role, to be able to provide local level training without a need for translation during training sessions.

### 3.3. HEWs Behavior Change 

HEWs reported increased readiness to change their own behaviors. Table 2 shows a significant change in readiness to make changes with clients at follow-up. (Table 2 here).

The frequency of use of four separate behavior change techniques was evaluated: (a) assessing prior knowledge, (b) discussing problems, (c) making plans, and (d) implementing changes with their clients. Data was collected before and one month after training, when the HEWs had the chance to try out the skills they had learned. The results are shown in Table 2. HEWs reported using all techniques more frequently at follow-up than before the training. It is important to note that baseline scores were already high (ceiling effect), suggesting that some HEWs considered they were already using these techniques before training.

### 3.4. Analysis of Video Recordings of Role Plays

A total of five video recordings were obtained from the workshops (Groups 1 and 2), which were all translated from Amharic. Overall, HEWs demonstrated the use of communication skills, including open questions, agenda-setting, and using reflective listening and BCTs of action plans, coping plans (problem-solving), social support, and prompting, shown in Table 3. 

These skills were not demonstrated by all HEWs in the role plays. In some cases, HEWs did not demonstrate the use of the communication skills from the training, using a didactic provision of information, for example: 

HEW:
*“Do you have a chicken at home and how frequently do you feed your children?”*


Mother:
*“yes”*


HEW:
*“are you giving eggs for your child?”*


Mother:
*“No, I used to sell eggs to buy another items and biscuit for my child.”*


HEW:
*“don’t sell all eggs to market, you have to give one egg for your child daily.”*


Other barriers to behavior change that the HEWs faced during their normal practice were evident during the role plays. These include reported financial pressures and social norms, as shown in the examples below:

HEW:
*“how are you working with our last time discussion to give one egg for your child every day?”*


Role Play Mother:
*“Yes, we have discussed last time to give one egg for my child everyday but I was not able to perform that. I used to sell the eggs to compensate the money for the purchase of other items for my home.”*


Role Play Mother:
*“in addition, my neighbours are not giving egg for their children”.*


## 4. Discussion

This project successfully adapted a previous behavior change program delivered in the UK [17] and used local research evidence and knowledge to develop and deliver a program of training in communication skills and behavior change to a group of HEWs in Ethiopia. The program aimed to improve infant nutrition in areas where this was a significant health problem. The training was pragmatically evaluated using a range of methods to ensure reliability as far as possible in this context.

The training intervention was very positively received overall, was acceptable and had a positive impact on HEW’s skills and knowledge about changing nutritional behavior in their communities. Some HEWs demonstrated good use of both the communication and behavior change techniques in simulated conversations with women, and overall, there was a willingness to learn and apply these techniques and an appreciation of their value. 

This experience in Ethiopia offered a great deal of learning about the factors that might influence the delivery and evaluation of training, and the need to ensure the future sustainability of training through, for example, mentoring. This training and evaluation contributes towards the translation of evidence-based behavior change science into health care practices [28] and will help to inform future delivery of communication and behavior change training in Ethiopia and in similar contexts. In contrast to similar projects elsewhere [29,30], this project was able to demonstrate HEWs’ effective use of behavior change techniques in a challenging context. It is important for this type of training which involves new concepts and skills, to be ongoing and sustainable, to allow HEWs to develop confidence, have the opportunity to practice, and normalize new behaviors. Including communication and behavior change skills in training allows HEWs to use their existing knowledge and skills of nutrition and health care to support women to greater effect by enhancing the collaborative way they work with women, enhancing their job satisfaction, and improves the problem-solving skills of the women themselves. 

There were several aspects of the program that worked very well, and there were some challenges that could be addressed in future developments of this or other projects to maximize their impact. 

### 4.1. Strengths 

Few similar studies have successfully delivered tailored interventions to a large sample of community health workers using evidence-based behavior change skills in this type of context. In our study, prior local qualitative research [9] helped to define specific cultural factors, existing nutritional behaviors, and HEW practices which informed the focus of training. 

The researchers had extensive experience of health behavior change training [17] and previous experience of working in low and middle-income countries. They were able to use approaches such as incorporating examples from local contexts into training and collaborating with ‘on the ground’ experts for adapting programs to local contexts working collaboratively to adapt the program at both a surface and deeper structural level [31,32]. The intervention was evidence-based but also very interactive, adaptable, and robust. Prior training experience and having an adaptable program made the modification of the program possible to account for increased numbers of HEW participants. Original plans were for 18 participants at each training event but many more attended. Collaboration with WFP staff nationally/locally, an experienced and knowledgeable translator, an informative field visit, and feedback from the HEWs themselves provided invaluable practical, contextual, and culturally relevant information, which enabled successful adaptation of a UK-developed model of training to a different context. 

### 4.2. Challenges

This type of training involving skills development is perhaps more suitable for smaller groups of participants to gain the most benefit by providing individual feedback, tailoring skills practice to individual needs, and small group interactions (for example, using role-play). Communication of complex psychological ideas/new ways of thinking in a different language was challenging, and it was difficult to pick up nuances of language and thought through simultaneous translation. Ideas and assumptions underpinning behavior change skills developed in a ‘western’ context require adaptation in order to have a good fit with other cultural environments. There are many challenges to successful cultural adaptation of psychosocial interventions, including tensions between fidelity and adaptation, applicability in different contexts, intra-cultural variations, and difficulties demonstrating evidence of effectiveness [30,31,32]. Nevertheless, it is important to develop rigorous processes for effective adaptation [31]. 

Accurate evaluation of new methods and approaches is difficult, and we had to rely on HEWs’ self-report of changes, which may have been inaccurate, with a tendency towards positive reporting due to the novelty of the approach. We cannot rule out reporting biases and identified ‘ceiling’ effects (i.e., a tendency to report positively). Ensuring robustness and validity in measurement using methods and scales unfamiliar to participants is an important but challenging consideration.

## 5. Conclusions 

This novel, theory-based mixed-method intervention showed positive improvements in behavior change and communication skills in a large cohort of HEWs in Ethiopia. It also identified important pointers for future work in similar contexts. It would be helpful to identify key messages (e.g., ‘protein is important for children’s growth and development’) as well as specific behavioral targets (‘feeding eggs to my child’) that fit with social norms and expectations of behavior within the cultural setting and are important to improve nutritional outcomes. It is also important to reinforce these messages by gaining endorsement from key individuals (e.g., husbands, grandmothers, and religious leaders) with influence in those communities. 

To communicate complex ideas around motivation and behavior, it may be better to focus intensive training on existing English speakers (translators with relevant experience, such as nurses) who can develop a shared understanding and disseminate more effectively to HEWs and who can continue to operate a system of mentoring and support. It is important to support future training with sets of sustainable materials for use in the field and to ensure consistency with future training and coaching. These could consist of video clips demonstrating behavior change and communication techniques, worksheets, cue cards, or informational scripts. These would need to be augmented with refresher training. Finally, for sustainability, it is important to develop a strong mentoring network consisting of individuals who are competent in the use of behavior change skills. 

## Figures and Tables

**Table 1 nutrients-13-01995-t001:** Communication skills and MAP Behavior Change Techniques used in the training.

Technique:	Definition
(a) Communication Skills:	
Open Questions	Questions that require an elaborated (rather than one or two word) response
Reflective Listening	Rephrase a statement to capture the implicit meaning and show understanding
Agenda setting	Agreeing focus based on what is most important to the person
Affirmations	Compliments or statements of appreciation and understanding
(b) MAP techniques	
Agreeing on outcome goal	Agree on a goal defined in terms of a positive outcome of desired behavior
Goal setting	Identify and set behavioral goals
Action Planning	Make a detailed plan of what the client will do, including as a minimum, when and where to act
Coping Planning (problem-solving)	Identify and plan ways of overcoming (specified) barriers
Social Support	Provide and/or identify sources of non-specific social support (or emotional support, or instrumental support)
Prompting (through another individual or an environmental prompt)	Identify a stimulus that elicits behavior (including personal prompts or external reminders)

**Table 2 nutrients-13-01995-t002:** Changes to HEW’s use of Behavior Change Techniques Post-Intervention.

Item	Pre-Workshop Mean (SD)	Follow-Up Mean (SD)	Significance t (df 66)	95%CI Mean Difference
Readiness ^a^	4.2 (0.90)	4.5 (0.56)	2.34, *p* = 0.02	0.04–0.46
Knowledge	7.4 (3.0)	9.6 (1.1)	5.65, *p* < 0.001	1.45–3.03
Problems	7.3 (2.6)	9.2 (1.4)	−5.38, *p* < 0.001	1.15–2.51
Planning	8.4 (2.3)	9.7 (1.1)	−4.11, *p* < 0.001	0.69–1.98
Make behavior changes	7.3 (2.7)	8.8 (1.7)	−3.95, *p* < 0.001	0.74–2.24

^a^ Score range 1–5.

**Table 3 nutrients-13-01995-t003:** Summary of Communication Skills and Behavior Change Techniques (BCTs) Observed in Video Recordings of Training Role-Plays.

Communication Skill or BCT	HEW Example ^a^
Setting the agenda (and goal setting) using an open question	‘How are things working to give your child one egg for your child every day?’
Action Planning and Prompts	‘When do you plan to buy the chicken? Who will remind you to go and buy the chicken or to go and buy the eggs?’
Coping Planning Exploring barriers (problem-solving)	‘When we discussed last time, you told me that you were not giving vegetables to your child. Can you tell what was your reason not to give vegetables to your child?’
Coping Planning Problem-solving	‘An example from my (your) neighbor on alternative options to provide fruit and vegetable for your children, which is selling other cereal or produce from your backyard.’
Problem-solving	HEW^a^: Shares experience of her neighbors and gives ideas of different options on how to prepare the food and feed the child. She gives ideas of instructing her daughter to feed the child. She gives examples of different people feeding the children one egg per day. An example of an ill mother who stays at home and takes responsibility for feeding the child one egg per day, and also her husband is taking on this responsibility. She asks: ‘which alternative is the best for you?’
Social support and prompts	‘Who will remind you (.. to give your child an egg)?’
Affirmations	‘You are a good mother’.
Review behavior and goal	‘I am so happy and I am sure you will do this in the future. When I return, I will check it.’

^a^ Reported verbatim from translation.

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
