# Peer review of "Enhancing Behavior Change Skills in Health Extension Workers in Ethiopia: Evaluation of an Intervention to Improve Maternal and Infant Nutrition"

_nutrients, 2021, doi:10.3390/nu13061995_

Round 1

Reviewer 1 Report

This is a very interesting paper concerning the effect of delivering a training intervention to  health extension workers  in Ethiopia’s primary health care system, in order to improve maternal and infant nutrition. The study is well designed and results are clearly exposed. The findings of the research appear to be very relevant to field of training interventions in LMIC contexts. 

Author Response

Thank you for the comments. This reviewers report does not require any response

Reviewer 2 Report

The undertaken and presented topic is very interesting, especially in the context of ethnic differences. But the work requires a lot of corrections and additions.

The aim of the work is too extensive, which makes it difficult to understand. The description of The Change Exchange should be moved up to the Introduction section and described in detail.

line 120-121 - please write, what kind of materials did you use?

line 136-142- After the first work shop, the program was slightly modified. Did the modification affect the whole cohort? were the conditions for the intervention still comparable for all participants?

line 147 - the participants section should describe how many women participated in the workshop in total. How many people directly attended each meeting, what was its duration. Please also describe why there were fewer meetings during the 2nd workshop.

Materials and methods are to be completely supplemented. It is necessary to provide the number of participants in the study and describe the materials used for the research.

Table 1 shows the differences in the results between the 2 stages of the study. Why the authors did not provide any variable characterizing the participants - e.g. age, having a child, merital status, education. It would be very interesting. Do the authors have such data and can present them in the form of a separate table?

Discussion - the section has become a research description, there is no reference to research by other authors. If there were no similar studies, it should be emphasized as the strength of the described study. The discussion section should be redrafted, a discussion should be held, and the strengths and weaknesses of the study should be indicated.

line 379-385 - this fragment should be moved to the discussion section.

Conclusion is too long and does not show exactly what results were achieved.

Author Response

The undertaken and presented topic is very interesting, especially in the context of ethnic differences. But the work requires a lot of corrections and additions.

The aim of the work is too extensive, which makes it difficult to understand. The description of The Change Exchange should be moved up to the Introduction section and described in detail.

Response:  It wasn't clear if the comment about the aim being 'too extensive' referred to the overall aim of the paper, or the specific aim stated on page 3, (now line 111).  We have made some overall  changes to the paper which hopefully makes it easier for the reviewer to understand in relation to the aims.  We have moved the description of The Change  Exchange to the introduction (now lines 85-91) as suggested. 

line 120-121 - please write, what kind of materials did you use?

Response: We have now listed the materials used under 'Intervention Delivery' on page 4, lines 228-231. 

line 136-142- After the first work shop, the program was slightly modified. Did the modification affect the whole cohort? were the conditions for the intervention still comparable for all participants?

Response: This has been clarified on Pages 3 and  4, lines 157-213. 

line 147 - the participants section should describe how many women participated in the workshop in total. How many people directly attended each meeting, what was its duration. Please also describe why there were fewer meetings during the 2nd workshop.

Response: We have clarified the number of workshops and attenders throughout the methods section of the document.  Attenders are labelled 'Group 1' and 'Group 2' to denote whether they attended the training on the first or second weekend delivery: e.g.  Page 3, lines 157-210; Page 4 lines 234-236. There were the same number of meetings for both workshops - Page 4, lines 219-223. 

Materials and methods are to be completely supplemented. It is necessary to provide the number of participants in the study and describe the materials used for the research.

Response: This has now been covered as above. Apologies for any confusion. 

Table 1 shows the differences in the results between the 2 stages of the study. Why the authors did not provide any variable characterizing the participants - e.g. age, having a child, merital status, education. It would be very interesting. Do the authors have such data and can present them in the form of a separate table?

Response: We did not collect data on individual characteristics of the HEWs in the workshops.  Since this was not essential for the study, and might have made participants personally identifiable, we felt it would not have been ethical to collect this type of personal data. 

Discussion - the section has become a research description, there is no reference to research by other authors. If there were no similar studies, it should be emphasized as the strength of the described study. The discussion section should be redrafted, a discussion should be held, and the strengths and weaknesses of the study should be indicated.

Response:  We have noted the reviewer's comments on the discussion.  As they say above, there are not relevant comparator studies available to allow us to compare outcome data from our study.  We have inserted a new reference [30] which relates to one study but in a different country with community workers in urban environments, so is not directly comparable Page 9, lines 452-454. We have included a sentence to this effect, Page 9, line 466. 

We had originally labelled our discussion sections 'Positive Factors' and 'Challenges' rather than 'Strengths' and 'Weaknesses'.  We have now changed 'Positive Factors'  to 'Strengths' but felt that 'Weaknesses' did not fully capture what we wanted to say in this section, so have retained the 'Challenges' sub-heading. 

line 379-385 - this fragment should be moved to the discussion section.

Response: This has been moved to lines 453-459 on page 9. 

Conclusion is too long and does not show exactly what results were achieved.

Response: This is now shorter, and includes a statement summarising the results. Page 10, Lines 516-518

Reviewer 3 Report

This is a much needed well designed study implemented in LMIC. Even that the sample size is small this study is valuable for its design and implementation.

Author Response

This is a much needed well designed study implemented in LMIC. Even that the sample size is small this study is valuable for its design and implementation.

Response: Thank you for the positive comments.  We felt that the sample size for this study was sufficient in comparison with others that have offered relatively complex theory-based, skills development training.  Normally this would be offered in small groups. 

Round 2

Reviewer 2 Report

the work was corrected, the authors coped very well with the revision of the manuscript, which gained substantial value. The work is interesting and can be published in this form.